# A Bayesian Nonparametric Topic Model with Variational Auto-Encoders

## Abstract

Topic modeling of text documents is one of the most important tasks in representation learning. In this work, we propose iTM-VAE, which is a Bayesian nonparametric (BNP) topic model with variational auto-encoders. On one hand, as a BNP topic model, iTM-VAE potentially has infinite topics and can adapt the topic number to data automatically. On the other hand, different with the other BNP topic models, the inference of iTM-VAE is modeled by neural networks, which has rich representation capacity and can be computed in a simple feed-forward manner. Two variants of iTM-VAE are also proposed in this paper, where iTM-VAE-Prod models the generative process in products-of-experts fashion for better performance and iTM-VAE-G places a prior over the concentration parameter such that the model can adapt a suitable concentration parameter to data automatically. Experimental results on 20News and Reuters RCV1-V2 datasets show that the proposed models outperform the state-of-the-arts in terms of perplexity, topic coherence and document retrieval tasks. Moreover, the ability of adjusting the concentration parameter to data is also confirmed by experiments.

## 1 Introduction

Probabilistic topic models focus on discovering the abstract "topics" that occur in a collection of documents and representing a document as a weighted mixture of the discovered topics. Classical topic models, the most popular being LDA (Blei et al., 2003), have achieved success in a range of applications, such as information retrieval (Wei & Croft, 2006), document understanding (Blei et al., 2003), computer vision (Rasiwasia & Vasconcelos, 2013) and bioinformatics (Rogers et al., 2005). A major challenge of topic models is that the inference of the distribution over topics does not have a closed-form solution and must be approximated, using either MCMC sampling or variational inference. Hence, any small change to the model requires re-designing a new inference method tailored for it. Moreover, as the model grows more expressive, the inference becomes increasingly complex, which becomes the bottleneck to discover the latent semantic structures of complicated data. Hence, black-box inference methods (Ranganath et al., 2014; Mnih & Gregor, 2014; Kingma & Welling, 2014b; Rezende et al., 2014), which require only limited knowledge from the models and can be flexibly applied to new models, is desirable for topic models.

Among all the black-box inference methods, Auto-Encoding Variational Bayes (AEVB) (Kingma & Welling, 2014b; Rezende et al., 2014) is a promising one for topic models. AEVB contains an inference network that can map a document directly to a variational posterior without the need for further local variational updates on test data, and the Stochastic Gradient Variational Bayes (SGVB) estimator allows efficient approximate inference for a broad class of posteriors, which makes topic models more flexible. Hence, an increasing number of work has been proposed recently to combine topic models with AEVB, such as (Miao et al., 2016; Srivastava & Sutton, 2017; Card et al., 2017; Miao et al., 2017).

Deciding the number of topics is another challenge for topic models. One option is to use model selection, which trains models with different topic numbers and selects the best on the validation set. Bayesian nonparametric (BNP) topic models, however, side-step this issue by making the number of topics adaptive to data. For example, Teh et al. (2006) proposed Hierarchical Dirichlet Process (HDP), which models each document with a Dirichlet Process (DP) and all DPs for documents in a corpus share a base distribution that is itself also from a DP. HDP extends LDA in that it can

adapt the number of topics to data. Hence, HDP has potentially an *infinite* number of topics and allows the number to grow as more documents are observed. Unlike the black-box inference based models, traditionally, one needs to redesign the inference methods when there are some changes in the generative process of HDP (Teh et al., 2006; Wang et al., 2011; Hughes et al., 2015).

In this work, we make progress on this problem by proposing an *infinite Topic Model with Variational Auto-Encoders* (iTM-VAE), which is a Bayesian nonparametric topic model with AEVB. Coupling Bayesian nonparametric techniques with deep neural networks, iTM-VAE is able to capture the uncertainty regarding to the number of topics, and the inference can be conducted in a simple feed-forward manner. More specifically, iTM-VAE uses a stick-breaking process (Sethuraman, 1994) to generate the mixture weights for a countably infinite set of topics, and use neural networks to approximate the variational posteriors. The main contributions of the paper are:

- We propose iTM-VAE, which, to our best knowledge, is the first Bayesian nonparametric topic model equipped with AEVB.

- We propose iTM-VAE-Prod whose distribution over words is a product of experts rather than a mixture of multinomials.

- We propose iTM-VAE-G, which helps the model to adjust the concentration parameter to data automatically.

- The experimental results show that iTM-VAE and its two variants outperform the state-of-the-art models on two challenging benchmarks significantly.

## 2    RELATED WORK

Topic models have been studied extensively in a variety of applications such as document modeling, information retrieval, computer vision and bioinformatics (Blei et al., 2003; Wei & Croft, 2006; Putthividhy et al., 2010; Rasiwasia & Vasconcelos, 2013; Rogers et al., 2005). Please see (Blei, 2012) for an overview. Recently, with the impressive success of deep learning, neural topic models have been proposed and achieved encouraging performance in document modeling tasks, such as Replicated Softmax (Hinton & Salakhutdinov, 2009), DocNADE (Larochelle & Lauly, 2012), fDARN (Mnih & Gregor, 2014) and NVDM (Miao et al., 2016). These models achieved competitive performance on modeling documents. However, they do not explicitly model the generative story of documents, hence are less explainable.

Several recent work has been proposed to model the generative procedure explicitly and the inference of the topic distributions is computed by deep neural networks. This makes these models interpretable, powerful and easily extendable. For example, Srivastava & Sutton (2017) proposed AVITM model, which embeds the original LDA (Blei et al., 2003) formulation with AEVB. By utilizing Laplacian approximation for the Dirichlet distribution, AVITM can be optimized by the Stochastic Gradient Variational Bayes (SGVB) (Kingma & Welling, 2014b; Rezende et al., 2014) estimator efficiently. AVITM achieved the state-of-the-art performance on the topic coherence metric (Lau et al., 2014), which indicates the topics learned match closely to human judgment.

The Bayesian nonparametric topic models Teh et al. (2006); Kim & Sudderth (2011); Archambeau et al. (2015); Lim et al. (2016), potentially have infinite topic capacity and are able to adapt the topic number to data. However, we do not notice any topic models that combine BNP techniques with deep neural networks. Actually, Nalisnick & Smyth (2017) proposed Stick-Breaking VAE (SB-VAE), which is a Bayesian nonparametric version of traditional VAE with a stochastic dimensionality. However, iTM-VAE differs with SB-VAE in that it is a kind of *topic model* that models discrete text data. Furthermore, we also proposed iTM-VAE-G which places a prior on the stick-breaking process such that the model is able to adapt the concentration parameter to data. Another related work is (Miao et al., 2017), which proposed GSM, GSB, RSB and RSB-TF to model documents. The RSB-TF from (Miao et al., 2017), which uses a heuristic indicator to guide the growth of the topic numbers, also has the ability to adapt the topic number. However, the performance of RSB-TF does not match its complexity. Instead, iTM-VAE exploits Bayesian nonparametric techniques to decide the number of topics, which is much more elegant. And the performance of iTM-VAE also outperforms RSB-TF.

## 3 PRELIMINARY

In this section, we briefly describe the stick-breaking process (Sethuraman, 1994; Murphy, 2012), Variational Auto-Encoders (Kingma & Welling, 2014b; Rezende et al., 2014) and the Kumaraswamy distribution (Kumaraswamy, 1980), which are all essential for iTM-VAE.

### 3.1 THE STICK-BREAKING PROCESS

We first describe the stick-breaking process, which is used to model the mixture weights over the countably infinite topics for the generative procedure of iTM-VAE. More specifically, the stick-breaking process generates an infinite sequence of mixture weights $\boldsymbol{\pi} = \{\pi_k\}_{k=1}^{\infty}$ as follows:

$$\nu_k \sim \text{Beta}(1, \alpha) \qquad \pi_k = \nu_k \prod_{l=1}^{k-1}(1 - \nu_l) = \nu_k(1 - \sum_{l=1}^{k-1} \pi_l). \tag{1}$$

This is often denoted as $\boldsymbol{\pi} \sim \text{GEM}(\alpha)$, where GEM stands for Griffiths, Engen and Mc-Closkey (Ewens, 1990) and $\alpha$ is the *concentration* parameter. One can see that $\pi_k$ satisfies $0 \leq \pi_k \leq 1$ and $\sum_{k=1}^{\infty} \pi_k = 1$.

### 3.2 VARIATIONAL AUTO-ENCODERS

Variational Auto-Encoder (VAE) is among the most successful generative models recently. Specifically, it assumes that a sample $x$ is generated mathematically as follows:

$$\mathbf{z} \sim p(\mathbf{z}), \mathbf{x} \sim p_\phi(\mathbf{x}|\mathbf{z}) \tag{2}$$

where $p(\mathbf{z})$ is the prior of the latent representation $\mathbf{z}$ and $p_\phi(\mathbf{x}|\mathbf{z})$ is a conditional distribution with parameters $\phi$ to generate $\mathbf{x}$. The approximation to the posterior of the generative process, $q_\psi(\mathbf{z}|\mathbf{x})$, is parametrized by an inference neural network with parameters $\psi$. According to (Kingma & Welling, 2014b; Rezende et al., 2014), the Evidence Lower Bound (ELBO) of VAE can be written as:

$$\mathcal{L}(\mathbf{x}|\phi, \psi) = \mathbb{E}_{q_\psi(\mathbf{z}|\mathbf{x})} \left[\log p_\phi(\mathbf{x}|\mathbf{z})\right] - \text{KL}\left(q_\psi(\mathbf{z}|\mathbf{x})||p(\mathbf{z})\right) \tag{3}$$

where KL is the Kullback-Leibler divergence.

The parameters $\phi$ and $\psi$ can be optimized jointly by applying the Stochastic Gradient Variational Bayes (SGVB) (Kingma & Welling, 2014b) estimator and *Reparameterization Trick* (RT). An essential requirement of SGVB and RT is that the latent variable $\mathbf{z}$ can be represented in a *differentiable, non-centered parameterization* (DNCP) (Kingma & Welling, 2014a), which allows the gradients to go through the MC expectation. However, this is not always satisfied by the Beta prior of the stick-breaking process for iTM-VAE. Hence, we resort to the little known Kumaraswamy distribution (Kumaraswamy, 1980) to solve this problem.

### 3.3 KUMARASWAMY DISTRIBUTION

Kumaraswamy distribution (Kumaraswamy, 1980) is a continuous distribution, which is mathematically defined as:

$$\text{Kumaraswamy}(x; a, b) = abx^{a-1}(1 - x^a)^{b-1} \tag{4}$$

where $x \in [0, 1]$ and $a, b > 0$. The inverse cumulative distribution function (CDF) can be expressed in a simple and closed-form formulation, and samples from Kumaraswamy distribution can be drawn by:

$$x = (1 - u^{\frac{1}{b}})^{\frac{1}{a}}, \text{ where } u \sim \text{Uniform}(0, 1) \tag{5}$$

Kumaraswamy is similar to Beta distribution, yet more suitable to SGVB since it satisfies the DNCP requirement. Moreover, the KL divergence between Kumaraswamy and Beta can be closely approximated in closed-form. Hence, we use it to model the inference procedure of iTM-VAE.

## 4 THE MODEL

In this section, we describe iTM-VAE, a Bayesian nonparametric topic model with VAE. Specifically, we first describe the generative process of iTM-VAE in Section 4.1, and then the inference process is introduced in Section 4.2. After that, two variants of the model, iTM-VAE-Prod and iTM-VAE-G, are described in Section 4.3 and Section 4.4, respectively.

### 4.1 THE GENERATIVE PROCEDURE

The generative process of iTM-VAE is similar to the original VAE. The key difference is that the topic distribution (latent variable) $\boldsymbol{\pi} = \{\pi_k\}_{k=1}^{\infty}$ is drawn from the GEM distribution to make sure of $\sum_{k=1}^{\infty} \pi_k = 1$ and $0 \leq \pi_k \leq 1$. Specifically, we suppose each topic $\boldsymbol{\theta}_k = \sigma(\boldsymbol{\phi}_k)$ is a probability distribution over vocabulary, where $\boldsymbol{\phi}_k \in \mathbb{R}^V$ is the parameter of the topic-specific word distribution, $\sigma(\cdot)$ is the softmax function and $V$ is the vocabulary size. In iTM-VAE, there are unlimited number of topics and we denote $\Theta = \{\boldsymbol{\theta}_k\}_{k=1}^{\infty}$ and $\Phi = \{\boldsymbol{\phi}_k\}_{k=1}^{\infty}$ as the collections of these countably infinite topics and the corresponding parameters. The generation of a document by iTM-VAE can then be mathematically described as:

- Draw a topic distribution $\boldsymbol{\pi} \sim \text{GEM}(\alpha)$
- Then we get a distribution $G(\boldsymbol{\theta}; \boldsymbol{\pi}, \Theta) = \sum_{k=1}^{\infty} \pi_k \delta_{\boldsymbol{\theta}_k}(\boldsymbol{\theta})$
- For each word $w_i$ in the document: 1) draw a topic $\hat{\boldsymbol{\theta}}_i \sim G(\boldsymbol{\theta}; \boldsymbol{\pi}, \Theta)$; 2) $w_i \sim \text{Cat}(\hat{\boldsymbol{\theta}}_i)$

where $\alpha$ is a hyper-parameter, $\text{Cat}(\hat{\boldsymbol{\theta}}_i)$ is a categorical distribution parameterized by $\hat{\boldsymbol{\theta}}_i$, and $\delta_{\boldsymbol{\theta}_k}(\boldsymbol{\theta})$ is a discrete dirac function, which equals to 1 when $\boldsymbol{\theta} = \boldsymbol{\theta}_k$ and 0 otherwise.

Thus, the joint probability of a document with $N$ words $\mathbf{w}_{1:N} = \{w_i\}_{i=1}^N$, the topic distribution $\boldsymbol{\pi}$ and the sampled topics $\hat{\boldsymbol{\theta}}_{1:N} = \{\hat{\boldsymbol{\theta}}_i\}_{i=1}^N$ can be written as:

$$p(\mathbf{w}_{1:N}, \boldsymbol{\pi}, \hat{\boldsymbol{\theta}}_{1:N}|\alpha, \Theta) = p(\boldsymbol{\pi}|\alpha) \prod_{i=1}^{N} p(w_i|\hat{\boldsymbol{\theta}}_i)p(\hat{\boldsymbol{\theta}}_i|\boldsymbol{\pi}, \Theta) \tag{6}$$

where $p(\boldsymbol{\pi}|\alpha) = \text{GEM}(\alpha)$, $p(\boldsymbol{\theta}|\boldsymbol{\pi}, \Theta) = G(\boldsymbol{\theta}; \boldsymbol{\pi}, \Theta)$ and $p(w|\boldsymbol{\theta}) = \text{Cat}(\boldsymbol{\theta})$.

Similar to (Srivastava & Sutton, 2017), we collapse the variable $\hat{\boldsymbol{\theta}}_{1:N}$ and rewrite Equation 6 as:

$$p(\mathbf{w}_{1:N}, \boldsymbol{\pi}|\alpha, \Theta) = p(\boldsymbol{\pi}|\alpha) \prod_{i=1}^{N} p(w_i|\boldsymbol{\pi}, \Theta) \tag{7}$$

where $p(w_i|\boldsymbol{\pi}, \Theta) = \text{CAT}(\bar{\boldsymbol{\theta}})$ and $\bar{\boldsymbol{\theta}} = \sum_{k=1}^{\infty} \pi_k \boldsymbol{\theta}_k$.

In practice, following (Miao et al., 2017), we factorize the parameter $\boldsymbol{\phi}_k$ of topic $\boldsymbol{\theta}_k$ as $\boldsymbol{\phi}_k = \boldsymbol{t}_k W$ where $\boldsymbol{t}_k \in \mathbb{R}^H$ is the $k$-th topic factor vector, $W \in \mathbb{R}^{H \times V}$ is the word factor matrix and $H \in \mathbb{R}_+$ is the factor dimension. For simplicity, we still use $\Phi$ to denote all the parameters regarding to the generative procedure of iTM-VAE. Note that, although we parameterize the generative process with $\Phi$, iTM-VAE is still a *nonparametric* model, since it has potentially infinite model capacity, and can grow the number of parameters with the amount of training data.

Notably, different with traditional nonparametric Bayesian topic models, the topics in iTM-VAE are not drawn from a base distribution, but are treated as part of the parameters of the model and are optimized directly. This key difference indicates that there is no need to use an additional base distribution to generate the countably infinite candidate topics such that these topics are shared by different documents. Instead, the topic parameters of iTM-VAE are shared across all documents naturally.

### 4.2 THE INFERENCE PROCEDURE

In this section, we describe the inference process of iTM-VAE, i.e. how to draw $\boldsymbol{\pi}$ given a document $\mathbf{w}_{1:N}$. To elaborate, suppose $\boldsymbol{\nu} = [\nu_1, \nu_2, \dots, \nu_{K-1}]$ is a $K-1$ dimensional vector where $\nu_k$ is a

random variable sampled from a Kumaraswamy distribution $\kappa(\nu; a_k, b_k)$ parameterized by $a_k$ and $b_k$, iTM-VAE models the joint distribution $q_\psi(\boldsymbol{\nu}|\mathbf{w}_{1:N})$ as: [1]

$$[a_1, \ldots, a_{K-1}; b_1, \ldots, b_{K-1}] = g(\mathbf{w}_{1:N}; \psi) \tag{8}$$

$$q_\psi(\boldsymbol{\nu}|\mathbf{w}_{1:N}) = \prod_{k=1}^{K-1} \kappa(\nu_k; a_k, b_k) \tag{9}$$

where $g(\mathbf{w}_{1:N}; \psi)$ is a neural network with parameters $\psi$. Then, $\boldsymbol{\pi} = \{\pi_k\}_{k=1}^K$ can be drawn by:

$$\boldsymbol{\nu} \sim q_\psi(\boldsymbol{\nu}|\mathbf{w}_{1:N}) \tag{10}$$

$$\boldsymbol{\pi} = (\pi_1, \pi_2, \ldots, \pi_{K-1}, \pi_K) = \left(\nu_1, \nu_2(1-\nu_1), \ldots, \nu_{k-1}\prod_{l=1}^{K-2}(1-\nu_l), \prod_{l=1}^{K-1}(1-\nu_l)\right) \tag{11}$$

In the above procedure, we truncate the infinite sequence of mixture weights $\boldsymbol{\pi} = \{\pi_k\}_{k=1}^\infty$ by $K$ elements, and $\nu_K$ is always set to 1 to ensure $\sum_{k=1}^K \pi_k = 1$. Notably, as discussed in (Blei et al., 2006), the truncation of variational posterior does *not* indicate that we are using a finite dimensional prior, since we *never* truncate the GEM prior. Moreover, the truncation level $K$ is not part of the generative procedure specification. Hence, iTM-VAE still has the ability to model the uncertainty of the number of topics and adapt it to data. People can manage to use truncation-free posteriors in the model, however, as observed by Nalisnick & Smyth (2017), it does not work well. On the opposite, the truncated-fashion posterior of iTM-VAE is simple and works well in practice.

iTM-VAE can be optimized by maximizing the Evidence Lower Bound (ELBO):

$$\mathcal{L}(\mathbf{w}_{1:N}|\Phi, \psi) = \mathbb{E}_{q_\psi(\boldsymbol{\nu}|\mathbf{w}_{1:N})}[\log p(\mathbf{w}_{1:N}|\boldsymbol{\pi}, \Phi)] - \mathrm{KL}\left(q_\psi(\boldsymbol{\nu}|\mathbf{w}_{1:N})||p(\boldsymbol{\nu}|\alpha)\right) \tag{12}$$

where we replace $\Theta$ with $\Phi$ for $p(\mathbf{w}_{1:N}|\boldsymbol{\pi}, \Phi)$ to emphasize that $\Phi$ is the parameter to be optimized, and $p(\boldsymbol{\nu}|\alpha)$ is products of $K-1$ Beta$(1, \alpha)$ distributions according to Section 3.1. The details of the optimization can be found in Appendix 7.2.

## 4.3 iTM-VAE with Products of Experts

In Equation 7, $\bar{\boldsymbol{\theta}}$ is a mixture of multinomials. One drawback of this formulation is that it cannot make any predictions that are sharper than the distributions being mixed, pointed out by (Hinton & Salakhutdinov, 2009), which may result in some topics that are of poor quality and do not match well with human judgment. One solution to this issue is to replace the mixture of multinomials with a weighted product of experts which is able to make sharper predictions than any of the constituent experts (Hinton, 2006). We develop a products-of-experts version of iTM-VAE, which is referred as iTM-VAE-Prod. Specifically, we compute a mixed topic distribution $\hat{\boldsymbol{\theta}} = \sigma(\sum_{k=1}^\infty \pi_k \boldsymbol{\phi}_k)$ for each document, where $\pi_k$ is sampled from GEM$(\alpha)$, and then each word of the document is sampled from Cat$(\hat{\boldsymbol{\theta}})$. The benefit of the products-of-experts is demonstrated in Section 5.

## 4.4 Placing a Prior on the Concentration Parameter

In the generative process, the concentration parameter $\alpha$ of GEM$(\alpha)$ can have significant impact on the growth of number of topics. The larger the $\alpha$ is, the more "breaks" it will create, and consequently, more topics will be used. Hence, it is generally reasonable to consider placing a prior on $\alpha$ so that the model can adjust the concentration parameter to data automatically.

Concretely, since the Gamma distribution is conjugate to Beta$(1, \alpha)$, we place a Gamma$(s_1, s_2)$ prior on $\alpha$. Then the ELBO of iTM-VAE can be written as:

$$\mathcal{L}(\mathbf{w}_{1:N}|\Phi, \psi) = \mathbb{E}_{q_\psi(\boldsymbol{\nu}|\mathbf{w}_{1:N})}[\log p(\mathbf{w}_{1:N}|\boldsymbol{\pi}, \Phi)] + \mathbb{E}_{q_\psi(\boldsymbol{\nu}|\mathbf{w}_{1:N})q(\alpha|\gamma_1, \gamma_2)}[\log p(\boldsymbol{\nu}|\alpha)]$$
$$- \mathbb{E}_{q_\psi(\boldsymbol{\nu}|\mathbf{w}_{1:N})}[\log q_\psi(\boldsymbol{\nu}|\mathbf{w}_{1:N})] - \mathrm{KL}(q(\alpha|\gamma_1, \gamma_2)||p(\alpha|s_1, s_2)) \tag{13}$$

---

[1]Ideally, Beta distribution is the most suitable probability candidate, since iTM-VAE assumes $\boldsymbol{\pi}$ is drawn from a GEM distribution in the generative process. However, as Beta does not satisfy the DNCP requirement of SGVB, we use the Kumaraswamy distribution, which is described in Section 3.3, for iTM-VAE.

where $p(\alpha|s_1, s_2) = \text{Gamma}(s_1, s_2)$, $p(v_k|\alpha) = \text{Beta}(1, \alpha)$, $q(\alpha|\gamma_1, \gamma_2)$ is the variational posterior for $\alpha$ with $\gamma_1, \gamma_2$ as parameters across the whole dataset. The derivation for Equation 13 can be found in Appendix 7.3. In experiments, we find iTM-VAE-Prod always performs better than iTM-VAE, therefore we only place the prior for iTM-VAE-Prod, and refer this variant as iTM-VAE-G.

## 5 EXPERIMENTS

In this section, we evaluate the performance of iTM-VAE and its variants on two public benchmarks: 20News and RCV1-V2. To make a fair comparison, we use exactly the same data and vocabulary as (Srivastava & Sutton, 2017). [2] We compare iTM-VAE and its variants with several state-of-the-arts, such as DocNADE, NVDM, NVLDA and ProdLDA (Srivastava & Sutton, 2017), GSM, GSB, RSB and RSB-TF, as well as some classical topic models such as LDA and HDP.

The configuration of the experiments is as follows. We use a two-layer fully-connected neural network for $g(\mathbf{w}_{1:N}; \psi)$ of Equation 8, and the number of hidden units is set to $256$ and $512$ for 20News and RCV1-V2, respectively. The factor dimension is set to $200$ and $1000$ for 20News and RCV1-V2, respectively. The truncation level $K$ in Equation 11 is set to $200$ so that the maximum topic numbers will never exceed the ones used by baselines. [3] Batch-Normalization (Ioffe & Szegedy, 2015) is used to stabilize the training procedure. The hyper-parameter $\alpha$ for GEM distribution is cross-validated on validation set from $[10, 20, 30, 50, 100]$. We use Adam (Kingma & Ba, 2015) to optimize the model and the learning rate is set to $0.01$ for all experiments. The code of iTM-VAE and its variants is available at `http://anonymous`.

### 5.1 PERPLEXITY EVALUATION

Perplexity is widely used by topic models to measure the goodness-to-fit capability, which is defined as: $\exp(-\frac{1}{D}\sum_{d=1}^{D}\frac{1}{|\mathbf{w}^d|}\log p(\mathbf{w}^d))$, where $D$ is the number of documents , and $|\mathbf{w}^d|$ is the number of words in the $d$-th document $\mathbf{w}^d$. Following previous work, the variational lower bound is used to estimate the perplexity.

Table 1 shows the perplexity of different topic models on 20News and RCV1-V2 datasets. Among these baselines, RSB and GSM (Miao et al., 2017) achieved the lowest perplexities on 20News and RCV1-V2, which are 785 and 521, respectively. While the perplexities achieved by iTM-VAE-Prod on these two benchmarks are 769 and 508, respectively, which performs better than the state-of-the-art. Moreover, Figure 1-(a) demonstrates perplexities of finite topic models with different number of topics on 20News. We can see that, a suitable topic number of these models should be around 10 and 20. Interestingly but as expected, the number of effective topics[4] discovered by iTM-VAE is about 19, which indicates that the Bayesian nonparametric topic model, iTM-VAE, has the ability to determine a suitable number of topics automatically.

### 5.2 TOPIC COHERENCE EVALUATION

As the quality of the learned topics is not directly reflected by perplexity (Newman et al., 2010), topic coherence is designed to match the human judgment. As Lau et al. (2014) showed that *Normalized Pointwise Mutual Information* (NPMI) matches the human judgment most closely, we adopt it as the measurement of topic coherence , same as (Miao et al., 2017; Srivastava & Sutton, 2017).[5] Since the number of topics discovered by iTM-VAE is dynamic, we define the *Effective Topic* as the topic which becomes the top-1 significant topic of a training sample among the training set more than $\tau \times D$ times, where $D$ is the number of training samples and $\tau$ is a ratio, which is set to $0.5\%$ in our experiments. [6]

---

[2] Since there are no labels for the datasets provided by (Srivastava & Sutton, 2017) and (Miao et al., 2017), we used a version from (Srivastava et al., 2013) for document retrieval tasks, which contains label information.

[3] In these baselines, at most 200 topics are used. Please refer Table 1 for details.

[4] The definition of Effective Topic can be found in Section 5.2.

[5] We use the code provided by (Lau et al., 2014) at `https://github.com/jhlau/topic_interpretability/`

[6] This means the *ineffective* topics are the ones act as top-1 important topic less than $0.5\%$ times among all documents in the training set.

| Methods | Perplexity | | | | Coherence | | | |
|---|---|---|---|---|---|---|---|---|
| | 20News | | RCV1-V2 | | 20News | | RCV1-V2 | |
| #Topics | 50 | 200 | 50 | 200 | 50 | 200 | 50 | 200 |
| LDA† | 893 | 1015 | 1062 | 1058 | 0.131 | 0.112 | — | |
| DocNADE | 797 | 804 | 856 | 670 | 0.086 | 0.082 | 0.079 | 0.065 |
| HDP† | 937 | | 918 | | — | | — | |
| NVDM† | 837 | 873 | 717 | 588 | 0.186 | 0.157 | — | |
| NVLDA | 1078 | 993 | 791 | 797 | 0.162 | 0.133 | 0.153 | 0.172 |
| ProdLDA | 1009 | 989 | 780 | 788 | 0.236 | 0.217 | 0.252 | 0.179 |
| GSM† | 787 | 829 | 653 | 521 | 0.223 | 0.186 | — | |
| GSB† | 816 | 815 | 712 | 544 | 0.217 | 0.171 | — | |
| RSB† | 785 | 792 | 662 | 534 | 0.224 | 0.177 | — | |
| RSB-TF† | 788 | | 532 | | — | | — | |
| iTM-VAE | 882 | | 1124 | | 0.205 | | 0.218 | |
| iTM-VAE-Prod | **769** | | **508** | | **0.291** | | **0.3** | |

†: Taken from Miao et al. (2017), where LDA is based on (Hoffman et al., 2010) and HDP is based on (Wang et al., 2011).

Table 1: Comparison of perplexity (lower is better) and topic coherence comparison (higher is better) of different topic models on 20News and RCV1-V2 datasets. The symbol "−" indicates either the model fails to converge within 24 hours, or the original paper does not provide the corresponding values.

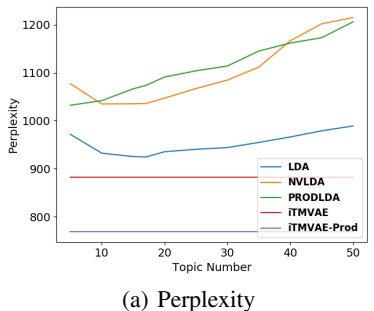

(a) Perplexity

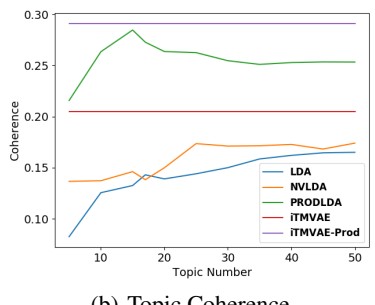

(b) Topic Coherence

Figure 1: Perplexity and topic coherence with different number of topics. The lines for iTM-VAE and iTM-VAE-Prod are horizontal as the topics used by the models are dynamic and adapted to data.

Following (Miao et al., 2017), we use an average over topic coherence computed by top-5 and top-10 words for all topics across five random runs, which is more stable and robust (Lau & Baldwin, 2016). Table 1 shows the topic coherence of different topic models on 20News and RCV1-V2 datasets. We can clearly see that iTM-VAE-Prod outperforms all the other topic models significantly, which indicates that the topics discovered by iTM-VAE-Prod match more closely to human judgment. Some topics learned by iTM-VAE-Prod are illustrated in Appendix 7.1.

We also illustrate the topic coherence of finite topic models with different numbers of topics, and compare them with iTM-VAE-Prod and iTM-VAE on 20News dataset, shown in Figure 1-(b). The topic coherence of iTM-VAE-Prod outperforms all baselines over all topic numbers. Another observation is that the best topic coherence of ProdLDA is achieved as the topic number is 15, which is close to the number of effective topics discovered by iTM-VAE-Prod.

## 5.3 Document Retrieval Evaluation

The document retrieval task is to evaluate the discriminative power of the document representations learned by each model. We compare iTM-VAE and iTM-VAE-Prod with LDA, DocNADE, NVLDA

and ProdLDA. [7] The setup of the retrieval task is as follows. The documents in the training/validation sets are used as the database for retrieval, and the test set is used as the query set. Given a query document, documents in the database are ranked according to the cosine similarities to the query. Precision/Recall (PR) curves can then be computed by comparing the label of the query with those of the database documents. For documents who have multiple labels (e.g. RCV1-V2), the PR curves for each of its labels are computed individually and then averaged for each query document. Finally, the global average of these curves is computed to compare the retrieval performance of each model.

Figure 2 illustrates the PR curves of different models with hidden representations of length 128. Specifically, the mean of the variational Gaussian posterior is used as the features for NVLDA and ProdLDA. Since the effective topic numbers of iTM-VAE-Prod and iTM-VAE are dynamic and usually much smaller than 128 on both datasets, we use a weighted sum of topic factor vectors over the topic distributions, where the factor dimension is 128. As shown in Figure 2(a) and Figure 2(b), iTM-VAE-Prod always yields competitive results on both datasets, and outperforms the others in most cases. We also map the latent representations learned by iTM-VAE-Prod to a 2D space by TSNE (Maaten & Hinton, 2008) and visualize the representations in Figure 2(c).

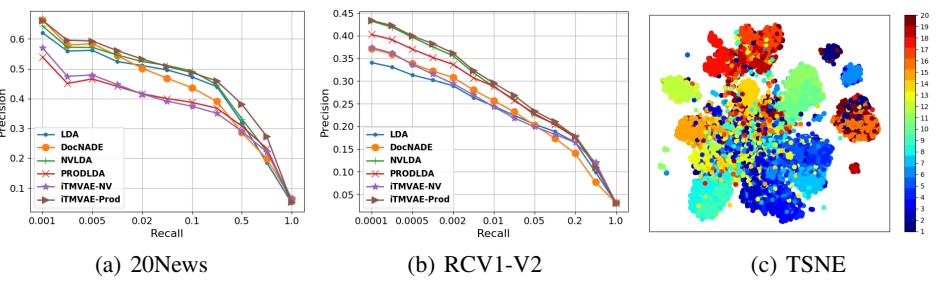

| (a) 20News | (b) RCV1-V2 | (c) TSNE |

Figure 2: Precision-Recall curves for document retrieval task on 20News (a) and RCV1-V2 (b). TSNE-visualization of the representations learned by iTM-VAE-Prod on 20News (c).

## 5.4 ADAPTING $\alpha$ AUTOMATICALLY BY iTM-VAE-G

As mentioned in Section 4.4, the concentration parameter $\alpha$ of GEM($\alpha$) has significant impact on the growth of the number of topics, which affects the performance of iTM-VAE significantly. As a result, $\alpha$ has to be chosen by cross-validation for better performance. Figure 3(a) also confirms that the number of effective topics increases with $\alpha$. Consequently, iTM-VAE-G, which places a Gamma($s_1, s_2$) prior on $\alpha$, is proposed to enable the model to adapt $\alpha$ to data automatically.

Another commonly mentioned problem of the optimization under VAE framework is that the latent representation will tend to collapse to the prior (Bowman et al.; Sønderby et al., 2016; Chen et al., 2017). In our model, this means the choice of $\alpha$ will control the number of the learned topics very tightly when the decoder is strong (e.g. iTM-VAE-Prod with large factor dimension $H$), which might cause the model to be lack of adaptive power. Common tricks to alleviate the training problem are annealing the relative weight of the KL divergence term (Sønderby et al., 2016), or regularizing the decoder (Bowman et al.). Rather than regularizing the decoder, iTM-VAE-G can be regarded as a relaxation on the prior placed on the latent space, which is more effective in improving the adaptive power of our model.

To verify this, we compare the adaptive power of iTM-VAE-Prod with KL annealing and decoder regularization, to iTM-VAE-G, on several subsets of 20News dataset, which contain 1, 2, 5, 10 and 20 (the whole dataset) classes, respectively. For these two models, we use a MLP of two layer of 256 units as the encoder, and the factor dimension of the decoder is $H = 200$. For iTM-VAE-Prod, we set $\alpha = 4$, and try different tricks to alleviate the collapse problem: KL annealing where the relative weight of the KL divergence term in the ELBO at epoch $n$ is $\min(0.005n, 1)$; Decoder regularization where the weight of the L2 regularization on the decoder is set to $0.1$. For iTM-VAE-G, we add a relatively non-informative prior Gamma($1, 0.25$) on $\alpha$, and initialize the global

---

[7]Since Miao et al. (2017) did not test the document retrieval performance and the code is not available now, we cannot compare with it during the draft preparation.

variational parameters $\gamma_1$ and $\gamma_2$ of Equation 13 the same as the non-informative prior. A SGD optimizer with a learning rate of $0.01$ is used to optimize $\gamma_1$ and $\gamma_2$. No KL annealing and decoder regularization is used for iTM-VAE-G.

The number of effective topics learned by iTM-VAE-Prod on subsets of 20News dataset is shown in Table 2. We can see that training tricks like KL-annealing and regularizing the decoder do not help much when the decoder is strong. However, by placing a prior on the concentration parameter $\alpha$, iTM-VAE-G can increase the adaptive power of the model.

The corpus-level variational posterior of $\alpha$ and the number of effective topics learned by iTM-VAE-G is shown in Table 3. As for iTM-VAE-G, before training, $\mathbb{E}_{q(\alpha|\gamma_1,\gamma_2)}[\alpha]$, the expectation of $\alpha$ given the variational posterior $q(\alpha|\gamma_1,\gamma_2)$ is 4. Once the training is done, $\mathbb{E}_{q(\alpha|\gamma_1,\gamma_2)}[\alpha]$ will be adjusted to the training set. Table 3 illustrates $\gamma_1$, $\gamma_2$, $\mathbb{E}_{q(\alpha|\gamma_1,\gamma_2)}[\alpha]$ and the number of effective topics that are learned from data. We can see that, if the training set contains only 1 class of documents, $\mathbb{E}_{q(\alpha|\gamma_1,\gamma_2)}[\alpha]$ will drop to 2.35, and only 6 effective topics are used to model the dataset. Whereas, when the training set consists of 10 classes of documents, $\mathbb{E}_{q(\alpha|\gamma_1,\gamma_2)}[\alpha]$ increases to 6.62, and 11 effective topics are discovered by the model to explain the dataset. This indicates that iTM-VAE-G can learn to adjust $\alpha$ to data.

Figure 3(b) illustrates the *topic coverage* w.r.t the number of topics when the training set contains 1, 2, 5, 10 and 20 classes, respectively. To this end, we compute the top-1 significant topic for each training document, and sort the topics according to the frequency that it is assigned as top-1. The topic coverage is then defined as the cumulative sum of these frequencies. Figure 3(b) shows that, with the increasing of the number of classes, more topics are utilized by iTM-VAE-G to reach the same level of topic coverage, which indicates that the model has the ability to adapt to data.

| #classes | $H = 200$ | $H = 200$ with KL annealing | $H = 200$ with decoder L2 regularization |
|---|---|---|---|
| 1 | 10 | 10 | 9 |
| 2 | 10 | 9 | 10 |
| 5 | 9 | 10 | 10 |
| 10 | 9 | 9 | 10 |
| 20 | 10 | 9 | 10 |

Table 2: Number of effective topics learned by iTM-VAE-Prod on subsets of 20News dataset.

| #classes | $\gamma_1$ | $\gamma_2$ | $\mathbb{E}_{q(\alpha|\gamma_1,\gamma_2)}[\alpha]$ | #Effective topics learned by iTM-VAE-G $H = 200$ |
|---|---|---|---|---|
| 1 | 18.30 | 7.80 | 2.35 | 6 |
| 2 | 25.03 | 6.38 | 3.92 | 9 |
| 5 | 34.10 | 6.19 | 5.51 | 9 |
| 10 | 42.96 | 6.49 | 6.62 | 11 |
| 20 | 52.97 | 7.23 | 7.33 | 14 |

Table 3: Learned posterior distribution of $\alpha$ and number of effective topics learned by iTM-VAE-G on subsets of 20News dataset.

## 6 CONCLUSION

In this paper, we propose iTM-VAE, which, to our best knowledge, is the first Bayesian nonparametric topic model that is modeled by Variational Auto-Encoders. Specifically, a stick-breaking prior is used to generate the mixture weights of countably infinite topics and the Kumaraswamy distribution is exploited such that the model can be optimized by AEVB algorithm. Two variants of iTM-VAE are also proposed in this work. One is iTM-VAE-Prod, which replaces the mixture of multinomials assumption of iTM-VAE with a product of experts for better performance. The other one is iTM-VAE-G which places a Gamma prior on the concentration parameter of the stick-breaking process such that the model can adapt the concentration parameter to data automatically. The advantage of iTM-VAE and its variants over the other Bayesian nonparametric topics models is that the inference

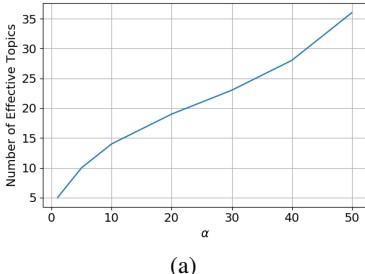 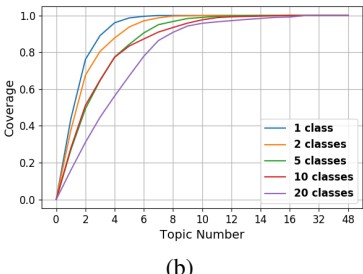

(a)  (b)

Figure 3: (a) Number of effective topics w.r.t $\alpha$ on 20News. (b) Topic coverage w.r.t number of topics learned by iTM-VAE-G.

is performed by feed-forward neural networks, which is of rich representation capacity and requires only limited knowledge of the data. Hence, it is flexible to incorporate more information sources to the model, and we leave it to future work. Experimental results on two public benchmarks show that iTM-VAE and its variants outperform the state-of-the-art baselines significantly.

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

# 7 APPENDIX

## 7.1 LEARNED TOPICS AND DISCUSSION

| | |
|---|---|
| **Geography** | turkish armenians turks armenia armenian turkey azerbaijan greek greece village |
| **Sports** | season team player nhl score playoff hockey game coach hitter |
| **Religion** | jesus bible god faith scripture christ doctrine belief eternal church |
| **Space** | orbit shuttle launch lunar spacecraft nasa satellite probe rocket moon |
| **Hardware** | scsi ide scsus motherboard ram controller upgrade meg cache floppy |
| **Encryption** | ripem escrow rsa des encrypt cipher privacy crypto chip nsa |
| **Trade** | shipping sale annual manual tor vs det cd price excellent |
| **X system** | window xterm font colormap server xlib widget xt windows toolkit |
| **Hockey**[†] | det tor buf cal pit que mon pt vs calgary |
| **Health** | msg[‡] patient disease symptom doctor food pain mouse cancer hospital |
| **Circuit** | voltage puck connector signal amp input circuit pin wire connect |
| **Lawsuit** | gun homicide militia weapon amendment handgun criminal firearm crime knife |
| **Traffic** | bike brake car tire ride engine honda bmw rear motorcycle |

†: All these words are about hockey teams of different cities, e.g. "que" means Quebec.

‡: "msg" means monosodium glutamate.

Table 4: Top 10 words of topics learned by iTM-VAE-Prod without cherry picking.

As shown in Table 4, iTM-VAE-Prod can learn topics that are diverse and of high quality. One possible reason is that the stick-breaking prior for the document-specific $\pi$ encourages the model to learn sparse representation, and the model can adjust the number of topics according to the data. Thus the topics can be sufficiently trained and of high diversity. The comparison of representation sparsity is illustrated in Figure 4(a).

In contrast, the topics learned by ProdLDA (Srivastava & Sutton, 2017) lack diversity. As we listed in Table 5, there are a lot of redundant topics. As a result, the latent representation learned by ProdLDA is of poor discriminative power. Figure 4(c) shows the TSNE-visualization of the representations learned by ProdLDA with the best topic coherence on 20News.

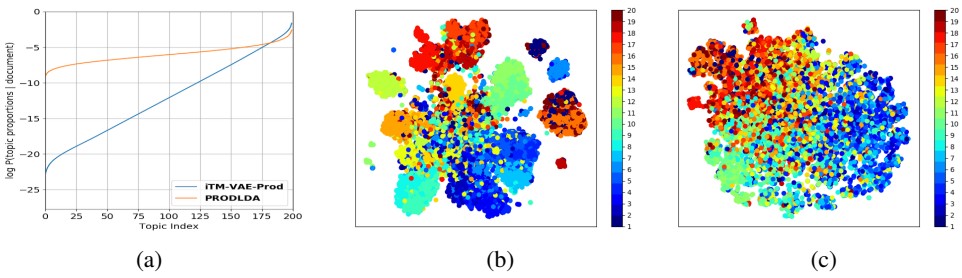

(a)          (b)          (c)

Figure 4: (a) Representation sparsity of different models on 20News. We sample one topic assignment $\pi$ for each document, sort and then average across the test set.[9](b) The TSNE-visualization of the representation learned by by iTM-VAE-Prod. (c) The TSNE-visualization of the representation learned by ProdLDA (Srivastava & Sutton, 2017) with the best topic coherence on 20News ($K = 50$).

## 7.2 THE EVIDENCE LOWER BOUND OF iTM-VAE

In this section we show how to compute the Evidence Lower Bound (ELBO) of iTM-VAE which can be written as:

$$\mathcal{L}(\mathbf{w}_{1:N}|\Phi,\psi) = \mathbb{E}_{q_\psi(\boldsymbol{\nu}|\mathbf{w}_{1:N})}\left[\log p(\mathbf{w}_{1:N}|\boldsymbol{\pi},\Phi)\right] - \mathrm{KL}\left(q_\psi(\boldsymbol{\nu}|\mathbf{w}_{1:N})||p(\boldsymbol{\nu}|\alpha)\right) \qquad (14)$$

[9]We use the code provided by (Srivastava & Sutton, 2017) to get their sparsity curve.

| Topics about Religion | 1 | jesus christian scripture faith god christ heaven christianity verse resurrection |
| | 2 | jesus christ doctrine revelation verse scripture satan christian interpretation god |
| | 3 | belief god passage scripture moral atheist christian truth principle jesus |
| | 4 | god belief existence faith jesus atheist bible christian religion sin |
| | 5 | jesus son holy christ father god doctrine heaven spirit prophet |
| | 6 | homosexual marriage belief islam moral christianity truth islamic religion god |
| Topics about Hardware | 7 | floppy controller scsus ide scsi ram hd mb cache isa |
| | 8 | printer meg adapter scsi motherboard windows modem mhz vga hd |
| | 9 | ide mb connector controller isa scsi scsus floppy jumper disk |
| | 10 | mb controller bio rom interface mhz scsus scsi floppy ide |
| | 11 | ide meg motherboard shipping adapter simm hd mhz monitor scsi |
| | 12 | ram controller dos bio windows disk scsi rom scsus meg |
| | 13 | honda motherboard bike amp quadra hd brake apple upgrade meg |
| Topics about Politics | 14 | decision stephanopoulos president armenian gay package congress myers february armenians |
| | 15 | armenian turkish genocide armenians turks jesus massacre muslim armenia muslims |
| | 16 | armenians father gang soldier neighbor apartment girl armenian troops rape |
| | 17 | muslim greek turks turkish armenian muslims village genocide armenia jews |
| | 18 | armenian turks armenians armenia turkish muslim massacre village turkey greek |
| | 19 | armenians armenian neighbor apartment woman soviet kill bus azerbaijan hide |
| Topics about Lawsuit | 20 | morality truth moral objective absolute belief murder existence principle human |
| | 21 | homicide vancouver seattle handgun firearm child states percent study file |
| | 22 | murder moral constitution morality criminal objective rights gun law weapon |
| . . . | . . . | . . . |

Table 5: Top 10 words of some redundant topics learned by ProdLDA.

- $\mathbb{E}_{q_\psi(\boldsymbol{\nu}|\mathbf{w}_{1:N})}\left[\log p(\mathbf{w}_{1:N}|\boldsymbol{\pi}, \Phi)\right]$:
  Similar to other VAE-based models, the SGVB estimator and *reparameterization* trick can be used to approximate this intractable expectation and propagate the gradient flow into the inference network $g$. Specifically, we have:

$$\mathbb{E}_{q_\psi(\boldsymbol{\nu}|\mathbf{w}_{1:N})}\left[\log p(\mathbf{w}_{1:N}|\boldsymbol{\pi}, \Phi)\right] = \frac{1}{L}\sum_{l=1}^{L}\sum_{i=1}^{N}\log p(w_i|\boldsymbol{\pi}^{(l)}, \Phi) \tag{15}$$

where $L$ is the number of Monte Carlo samples in the SGVB estimator and can be set to 1. $N$ is the number of words in the document.
According to Section 4.2, $\boldsymbol{\pi}^{(l)}$ can be obtained by

$$[a_1, \ldots, a_{K-1}; b_1, \ldots, b_{K-1}] = g(\mathbf{w}_{1:N}; \psi) \tag{16}$$

$$\nu_k \sim \kappa(\nu; a_k, b_k) \tag{17}$$

$$\boldsymbol{\pi} = (\pi_1, \pi_2, \ldots, \pi_{K-1}, \pi_K)$$
$$= \left(\nu_1, \nu_2(1-\nu_1), \ldots, \nu_{k-1}\prod_{l=1}^{K-2}(1-\nu_l), \prod_{l=1}^{K-1}(1-\nu_l)\right) \tag{18}$$

where $g(\mathbf{w}_{1:N}; \psi)$ is an inference network with parameters $\psi$, $\kappa$ denotes the Kumaraswamy distribution and $K \in \mathbb{R}_+$ is the truncation level. Here we omit the superscript $*^{(l)}$ for simplicity.
According to the generative procedure in Section 4.1, $p(w_i|\boldsymbol{\pi}^{(l)}, \Phi)$ can be computed by

$$p(w_i|\boldsymbol{\pi}^{(l)}, \Phi) = \begin{cases} \sum_{k=1}^{\infty}\pi_k^{(l)}\sigma(\boldsymbol{t}_k W) \approx \sum_{k=1}^{K}\pi_k^{(l)}\sigma(\boldsymbol{t}_k W) & \text{iTM-VAE} \\ \sigma(\sum_{k=1}^{\infty}\pi_k^{(l)}\boldsymbol{t}_k W) \approx \sigma(\sum_{k=1}^{K}\pi_k^{(l)}\boldsymbol{t}_k W) & \text{iTM-VAE-Prod} \end{cases} \tag{19}$$

where $\boldsymbol{t}_k \in \mathbb{R}^H$ is the $k$-th topic factor vector, $W \in \mathbb{R}^{H \times V}$ is the word factor matrix, $H$ is the factor dimension, $V$ is the vocabulary size and $\sigma(\cdot)$ is the softmax function.

- KL $(q_\psi(\boldsymbol{\nu}|\mathbf{w}_{1:N})||p(\boldsymbol{\nu}|\alpha))$: By applying the KL divergence of a Kumaraswamy distribution $\kappa(\nu; a_k, b_k)$ from a beta distribution $p(\nu; 1, \alpha)$, we have:

$$
\begin{aligned}
\mathrm{KL}\left(q_\psi(\boldsymbol{\nu}|\mathbf{w}_{1:N})||p(\boldsymbol{\nu}|\alpha)\right) &= \sum_{k=1}^{K-1} \mathrm{KL}\left(q_\psi(\nu_k|\mathbf{w}_{1:N})||p(\nu_k|\alpha)\right) \\
&= \sum_{k=1}^{K-1} \frac{a_k - 1}{a_k}\left(-\gamma - \Psi(b_k) - \frac{1}{b_k}\right) + \log a_k b_k + \log B(1, \alpha) \\
&\quad + (\alpha - 1)\sum_{m=1}^{\infty} \frac{b_k}{m + a_k b_k} B\left(\frac{m}{a_k}, b_k\right) - \frac{b_k - 1}{b_k}
\end{aligned}
\tag{20}
$$

where $B(\cdot)$ is the Beta function and $\gamma$ is the Euler's constant.

## 7.3 THE EVIDENCE LOWER BOUND OF ITM-VAE-G

In this section we show how to compute the Evidence Lower Bound (ELBO) of iTM-VAE-G which can be written as:

$$
\begin{aligned}
\mathcal{L}(\mathbf{w}_{1:N}|\Phi, \psi) = \ &\mathbb{E}_{q_\psi(\boldsymbol{\nu}|\mathbf{w}_{1:N})}[\log p(\mathbf{w}_{1:N}|\boldsymbol{\pi}, \Phi)] + \mathbb{E}_{q_\psi(\boldsymbol{\nu}|\mathbf{w}_{1:N})q(\alpha|\gamma_1, \gamma_2)}[\log p(\boldsymbol{\nu}|\alpha)] \\
&- \mathbb{E}_{q_\psi(\boldsymbol{\nu}|\mathbf{w}_{1:N})}[\log q_\psi(\boldsymbol{\nu}|\mathbf{w}_{1:N})] - \mathrm{KL}(q(\alpha|\gamma_1, \gamma_2)||p(\alpha|s_1, s_2))
\end{aligned}
\tag{21}
$$

Specifically, each item in Equation 21 can be obtained as follows:

- $\mathbb{E}_{q_\psi(\boldsymbol{\nu}|\mathbf{w}_{1:N})}[\log p(\mathbf{w}_{1:N}|\boldsymbol{\pi}, \Phi)]$:
  The derivation is exactly the same as Appendix 7.2.

- $\mathbb{E}_{q_\psi(\boldsymbol{\nu}|\mathbf{w}_{1:N})q(\alpha|\gamma_1, \gamma_2)}[\log p(\boldsymbol{\nu}|\alpha)]$:
  Recall that the prior of the stick length variable $\nu_k$ is Beta(1,$\alpha$): $p(v_k|\alpha) = \alpha(1 - \nu_k)^{\alpha-1}$ and the variational posterior of the concentration parameter $\alpha$ is a gamma distribution $q(\alpha; \gamma_1, \gamma_2)$, we have

$$
\begin{aligned}
\mathbb{E}_{q_\psi(\boldsymbol{\nu}|\mathbf{w}_{1:N})q(\alpha|\gamma_1, \gamma_2)}[\log p(\boldsymbol{\nu}|\alpha)] &= \mathbb{E}_{q_\psi(\boldsymbol{\nu}|\mathbf{w}_{1:N})}\Big[\sum_{k=1}^{K-1} \mathbb{E}_{q(\alpha|\gamma_1, \gamma_2)}[\log \alpha + (\alpha - 1)\log(1 - \nu_k)]\Big] \\
&= (K - 1)\mathbb{E}_{q(\alpha|\gamma_1, \gamma_2)}[\log \alpha] + \sum_{k=1}^{K-1} \frac{\gamma_1 - \gamma_2}{\gamma_2} \mathbb{E}_{q_\psi(\nu_k|\mathbf{w}_{1:N})}[\log(1 - \nu_k)]
\end{aligned}
\tag{22}
$$

Now, we provide more details about the calculation of these two expectations in Equation 22 as follows:

○ $\mathbb{E}_{q(\alpha|\gamma_1, \gamma_2)}[\log \alpha]$:
  Fisrt, we can write the gamma distribution $q(\alpha; \gamma_1, \gamma_2)$ in its exponential family form:

$$
q(\alpha; \gamma_1, \gamma_2) = \frac{1}{\alpha}\exp\big(-\gamma_2\alpha + \gamma_1\log\alpha - (\log\Gamma(\gamma_1) - \gamma_1\log\gamma_2)\big)
\tag{23}
$$

  Considering the general fact that the derivative of the log normalizor $\log\Gamma(\gamma_1) - \gamma_1\log\gamma_2$ of a exponential family distribution with respect to its natural parameter $\gamma_1$ is equal to the expectation of the sufficient statistic $\log\alpha$, we can compute $\mathbb{E}_{q(\alpha|\gamma_1, \gamma_2)}[\log\alpha]$ in the first term of Equation 22 as follows:

$$
\mathbb{E}_{q(\alpha|\gamma_1, \gamma_2)}[\log\alpha] = \Psi(\gamma_1) - \log\gamma_2
\tag{24}
$$

  where $\Psi$ is the digamma function, the first derivative of the log Gamma function.

○ $\mathbb{E}_{q_\psi(\nu_k|\mathbf{w}_{1:N})}[\log(1 - v_k)]$:
  By applying the Taylor expansion, $\mathbb{E}_{q(\nu_k|\mathbf{w}_{1:N})}[\log(1 - v_k)]$ can be written as the infinite sum of the Kumaraswamy's $m$th moment:

$$
\mathbb{E}_{q_\psi(\nu_k|\mathbf{w}_{1:N})}[\log(1 - v_k)] = -\sum_{m=1}^{\infty} \frac{1}{m}\mathbb{E}_{q_\psi(\nu_k|\mathbf{w}_{1:N})}[v_k^m] = -\sum_{m=1}^{\infty} \frac{b_k}{m + a_k b_k} B(\frac{m}{a_k}, b_k)
\tag{25}
$$

  where $B(\cdot)$ is the Beta function.

By substituting Equation 24 and Equation 25 into Equation 22, we can obtain:

$$\mathbb{E}_{q_\psi(\boldsymbol{\nu}|\mathbf{w}_{1:N})q(\alpha|\gamma_1,\gamma_2)}[\log p(\boldsymbol{\nu}|\alpha)]$$

$$= (K-1)(\Psi(\gamma_1) - \log \gamma_2) - \frac{\gamma_1 - \gamma_2}{\gamma_2} \sum_{k=1}^{K-1} \sum_{m=1}^{\infty} \frac{b_k}{m + a_k b_k} B(\frac{m}{a_k}, b_k)$$

(26)

- $-\mathbb{E}_{q_\psi(\boldsymbol{\nu}|\mathbf{w}_{1:N})}[\log q_\psi(\boldsymbol{\nu}|\mathbf{w}_{1:N})]$:
  According to Section 4.11 of (Michalowicz et al., 2013), the Kumaraswamy's entropy is given as

$$
\begin{aligned}
-\mathbb{E}_{q_\psi(\boldsymbol{\nu}|\mathbf{w}_{1:N})}[\log q_\psi(\boldsymbol{\nu}|\mathbf{w}_{1:N})] &= -\sum_{k=1}^{K-1} \mathbb{E}_{q_\psi(\nu_k|\mathbf{w}_{1:N})}[\log q_\psi(\nu_k|\mathbf{w}_{1:N})] \\
&= \sum_{k=1}^{K-1} -\log(a_k b_k) + \frac{a_k - 1}{a_k}(\gamma + \Psi(b_k) + \frac{1}{b_k}) + \frac{b_k - 1}{b_k}
\end{aligned}
$$

(27)

where $\gamma$ is the Euler's constant.

- $\mathrm{KL}(q(\alpha|\gamma_1, \gamma_2)||p(\alpha|s_1, s_2))$:
  The KL divergence of one gamma distribution $q(\alpha; \gamma_1, \gamma_2)$ from another gamma distribution $p(\alpha; s_1, s_2)$ evaluates to

$$\mathrm{KL}(q||p) = s_1 \log \frac{\gamma_2}{s_2} - \log \frac{\Gamma(\gamma_1)}{\Gamma(s_1)} + (\gamma_1 - s_1)\Psi(\gamma_1) - (\gamma_2 - s_2)\frac{\gamma_1}{\gamma_2}$$

(28)

