# OpenReview forum: "A Bayesian Nonparametric Topic Model with Variational Auto-Encoders"
_ICLR.cc/2018/Conference — Reject_

### Official Review · AnonReviewer3 · 2017-11-17
**well rounded work with good experimention and nice ideas, but some questions**

**Rating:** 7
**Confidence:** 4

**Review:**

"topic modeling of text documents one of most important tasks"
Does this claim have any backing?

"inference of HDP is more complicated and not easy to be applied to new models"  Really an artifact of the misguided nature of earlier work. The posterior for the $\vec\pi$ of a elements of DP or HDP can be made a Dirichlet, made finite by keeping a "remainder" term and appropriate augmentation.  Hughes, Kim and Sudderth (2015) have avoided stick-breaking and CRPs altogether, as have others in earlier work. Extensive models building on simple HDP doing all sorts of things have been developed.

Variational stick-breaking methods never seemed to have worked well.  I suspect you could achieve better results by replacing them as well, but you would have to replace the tree of betas and extend your Kumaraswamy distribution, so it may not work.  Anyway, perhaps an avenue for future work.

"infinite topic models" I've always taken the view that the use of the word "infinite" in machine learning is a kind of NIPSian machismo. In HDP-LDA at least, the major benefit in model performance comes from fitting what you call $\vec\pi$, which is uniform in vanilla LDA, and note that the number of topics "found" by a HDP-LDA sampler can be made to vary quite widely by varying what you call $\alpha$, so any statement about the "right" number of topics is questionable.  So the claim in 3rd paragraph of Section 2, "superior" and "self-determined topic number" I'd say are misguided.  Plenty of experimental work to support this.

In Related Work, you seem to only mention HDP for non-parametric topic models.  More work exists, for instance using Pitman-Yor distributions for modelling words and using Gibbs samplers that are efficient and don't rely on the memory hungry HCRP.

Good to see a prior is placed on the concentration parameter.  Very important and not well done in the community, usually.
ADDED:  Originally done by Teh et al for HDP-LDA, and subsequently done
by several, including Kim et al 2016.   Others stress the importance of this.  You need to
cite at least Teh et al. in 5.4 to show this isn't new and the importance is well known.

The Prod version is a very nice idea.  Great results.  This looks original, but I'm not expert enough in the huge masses of new deep neural network research popping up.

You've upped the standard a bit by doing good experimental work.  Oftentimes this is done poorly and one is left wondering.  A lot of effort went into this.
ADDED:   usually like to see more data sets experimented with

What code is used for HDP-LDA?  Teh's original Matlab HCRP sampler does pretty well because at least he samples hyperparameters and can scale to 100k documents (yes, I tried). The comparison with LDA makes me suspicious. For instance, on 20News, a good non-parametric LDA will find well over 400 topics and roundly beat LDA on just 50 or 200.  If reporting LDA, or HDP-LDA, it should be standard to do hyperparameter fitting and you need to mention what you did as this makes a big difference.
ADDED:   20News results still poor for HPD, but its probably the implementation used ... their
        online variational algorithm only has advantages for large data sets

Pros:
* interesting new prod model with good results
* alternative "deep" approach to a HDL-LDA model
* good(-ish) experimental work
Cons:
* could do with a competitive non-parametric LDA implementation

ADDED:   good review responses generally

---

> ### Author Response · Authors · 2017-12-06
> **Thanks for the review, and the questions are replied**
>
> We thank the reviewer for the insightful and valuable comments, and we will consider seriously the avenue of future work you point out to us. We address the concerns of the reviewer as follows.
>
> Q1. “Does this claim have any backing? ’Inference of HDP is more complicated and not easy to be applied to new models‘ …”
>
> Thanks very much for this comment. We agree with you that the posterior of DP or HDP can be made to a Dirichlet by keeping a remainder term and appropriate augmentation, which makes the model easier to be optimized. By saying “Inference of HDP is more complicated and not easy to be applied to new models  even with small changes in the generative process”, we mean that, unlike the black-box inference based models, people might need to redesign the inference methods when there are some changes in the generative process of HDP, which is not quite flexible. We have made this point clearer in the revision.
>
> Q2. “So the claim in 3rd paragraph of Section 2, "superior" and "self-determined topic number" I'd say are misguided. ”
>
> Thanks for the suggestion that makes our paper more rigorous. We have modified the sentence to “The Bayesian nonparametric topic models, such as HDP, potentially have infinite topic capacity and are able to adapt the topic number to data.”
>
> Q3.  “In Related Work, you seem to only mention HDP for non-parametric topic models.  More work exists, for instance using Pitman-Yor distributions for modelling words and using Gibbs samplers that are efficient and don't rely on the memory hungry HCRP.”
>
> Thanks for the suggestion!  We have added more references for nonparametric topic model in the related work of the revision, such as Kim & Sudderth (2011); Archambeau et al. (2015) and Lim et al. (2016).
>
> Q4. “What code is used for HDP-LDA? “
>
> Thanks for this comment. We take the results of HDP and LDA from Miao et al. (2017), since we use the exactly same datasets as Miao et al. (2017) (We are appreciated that Miao provides the exactly same datasets to us privately, hence we can take the results directly). According to Miao et al. (2017), the HDP is based on Wang et al. (2011), which is an online variational inference algorithm, and the LDA is based on the online variational inference model of Hoffman et al., (2010). The results of LDA in Figure 1 are also based on Hoffman et al., (2010), where we use the implementation from http://scikit-learn.org/stable/modules/generated/sklearn.decomposition.LatentDirichletAllocation.html.  We have made it clearer in the revision.

---

### Official Review · AnonReviewer2 · 2017-11-27
**The paper constructs infinite Topic Model with Variational Auto-Encoders by combining stick-breaking variational auto-encoder (SB-VAE) of Nalisnick & Smyth (2017) with latent Dirichlet allocation (LDA). The paper is not sufficiently novel and the proposed Bayesian nonparametric generative model is not theoretically sound.**

**Rating:** 3
**Confidence:** 4

**Review:**

The paper constructs infinite Topic Model with Variational Auto-Encoders (iTM-VAE) by combining stick-breaking variational auto-encoder (SB-VAE) of Nalisnick & Smyth (2017) with latent Dirichlet allocation (LDA) and several inference techniques used in Miao et al. (2016 & 2017). A main difference from Autoencoded Variational Inference For Topic Model (AVITM) of Srivastava & Sutton (2017), which had already applied VAE to LDA, is that the Dirichlet-distributed topic distribution vector for each document is now imposed with a stick-breaking prior. To address the challenge of reparameterizing the beta distributions used in stick-breaking, the paper follows SB-VAE to use the Kumaraswamy distributions to approximate the beta distributions.

The novelty of the paper does not appear to be significant, considering that most of the key techniques used in the paper had already appeared in several related papers, such as Nalisnick & Smyth (2017), Srivastava & Sutton (2017), and Miao et al. (2016 & 2017).

While experiments show the proposed models outperform the others quantitatively (perplexity and coherence), the paper does not provide sufficient justifications on why ITM-VAE is better. In particular, it provides little information about how the two baselines, LDA and HDP, are implemented (e.g., via mean-field variational inference, VAE, or MCMC?) and how their perplexities and topic coherences are computed. In addition, achieving the best performance with about 20 topics seem quite surprising for 20news and RCV-v2. It is hard to imagine 20 news, which consists of articles in 20 different newsgroups, can be well characterized by about 20 different topics. Is there a tuning parameter that significantly impacts the number of topics inferred by iTM-VAE?

Another clear problem of the paper is that the “Bayesian nonparametric” generative procedure specified in Section 4.1 is not correct in theory. More specifically, independently drawing the document-specific pi vectors from the stick-breaking processes will lead to zero sharing between the atoms of different stick-breaking process draws. To make the paper theoretically sound as a Bayesian nonparametric topic model that uses the stick-breaking construction, please refer to Teh et al. (2006, 2008) and Wang et al. (2011) for the correct construction that ties the document-specific pi vectors with a globally shared stick-breaking process.

---

> ### Author Response · Authors · 2017-12-06
> **[Part-3]**
>
> Q4. “In particular, it provides little information about how the two baselines, LDA and HDP, are implemented (e.g., via mean-field variational inference, VAE, or MCMC?) and how their perplexities and topic coherences are computed.”
>
> Thanks very much for this comment. We take the results of LDA and HDP from Miao et al. (2017), since we use the exactly same datasets as them (Yes, we are appreciated that Miao kindly provides the exactly same datasets to us privately.). According to Miao et al. (2017), they use Hoffman et al. , (2010) for LDA and Wang et al. (2011) for HDP, which are both based on variational inference. The LDA results in Figure 1 are also based on Hoffman et al. , (2010) (http://scikit-learn.org/stable/modules/generated/sklearn.decomposition.LatentDirichletAllocation.html).
> We have listed the equation for perplexity in Section 5.1, and we use the code provided by Lau et al (2014) (the code is at https://github.com/jhlau/topic_interpretability/ , which is also used by Srivastava & Sutton (2017)) to compute topic coherence. We make this point clearer in the revision.
>
> Q5. “It is hard to imagine 20 news, which consists of articles in 20 different newsgroups, can be well characterized by about 20 different topics.”
>
> Yes, it seems amazing that about 20 different topics can explain 20News dataset quite well. However, according to the description of 20News (http://qwone.com/~jason/20Newsgroups/), some of the newsgroups are very closely related to each other. Since 20News is a quite small dataset and many groups are very closely related (e.g. comp.graphics, comp.sys.mac.hardware, comp.windows.x), it is reasonable to explain the data with a small number of topics. Table 5 in the revision shows that among 50 topics learned by AVITM, there are many redundant topics. Moreover, the curves in Figure 1-(a) also confirms that about 20 topics is a good choice for the other topic models.
>
> Q6. “Is there a tuning parameter that significantly impacts the number of topics inferred by iTM-VAE?”
>
> The concentration parameter $\alpha$ is the one that significantly impacts the number of topics inferred by iTM-VAE. Since it is a very important parameter, we place a prior on $\alpha$ which helps the model to adjust  $\alpha$ to data automatically. Section 5.4 demonstrates the effectiveness of the prior. If we do not place the prior, iTM-VAE-Prod with strong decoder cannot adapt well to different sub-sampled subsets of 20News dataset. Common training techniques, e.g. KL annealing, decoder regularization, do not alleviate this problem significantly. While iTM-VAE-G has better adaptive ability w.r.t dataset size. This is also one of the key contributions of the paper.
>
> We hope that this rebuttal can address the concerns and some misunderstandings of the reviewer. Please let us know whether the reviewer has other comments. We are looking forward to hearing from you.

---

> ### Author Response · Authors · 2017-12-06
> **[Part-2]**
>
> Q2. “The novelty of the paper does not appear to be significant, considering that most of the key techniques used in the paper had already appeared in several related papers, such as Nalisnick & Smyth (2017), Srivastava & Sutton (2017), and Miao et al. (2016 & 2017). ”
>
>
> We respectfully beg to differ that the novelty of the paper is not significant.
> 1) Although the paper shares the similar motivation with Srivastava & Sutton (2017) that using neural network to do topic modeling tasks, the architecture of iTM-VAE is quite different with AVITM. Moreover, our model is a kind of nonparametric model while AVITM is not.  At last, the performance (perplexity, topic coherence) of our model is much better than AVITM.
>
> 2) Compared with Nalisnick & Smyth (2017), which proposes to replace the normal prior with a stick breaking prior for traditional VAE, our model is a kind of  topic model for discrete text data. Moreover, we propose to place a prior on the concentration parameter such that the model is able to adjust the concentration parameter to data automatically. As commented by AR3, this technique is “very important and not well done in the community, usually”.  We have demonstrated the adaptive power of iTM-VAE-G in Section 5.4.
>
> Furthermore, iTM-VAE-G can alleviate another commonly mentioned problem of the optimization under VAE framework:  the latent representation will tend to collapse to the prior, which might leads to poor adaptive power in our model if the decoder is strong.  iTM-VAE-G can increase the adaptive power of the model in an elegant way. Please refer to the newly added discussion in the second paragraph of Section 5.4 and Table 2 in the revision for details.
>
> 3) The common point of our model with Miao et al. (2017) is that both of them can adapt the number of topics to data automatically. Miao et al. (2017) uses a heuristic indicator to instruct the growing of the topic number. While in our model, the adaptation of topic number is done in a natural Bayesian way. Placing a prior and carrying out corpus-level variational inference on the concentration parameter is an elegant way of adapting model power to datasets.
>
> Please also refer to Q3 of AR1 for the discussion of the novelty. Thanks very much!
>
> Q3. “While experiments show the proposed models outperform the others quantitatively (perplexity and coherence), the paper does not provide sufficient justifications on why ITM-VAE is better.”
>
> Good question and thanks for this comment.  According to our observation, the advantage of iTM-VAE lies in that the model adjusts the number of topics according to the data and the stick-breaking prior encourages the the model to learn sparse representations. Hence, the topics learned by iTM-VAE are usually diverse and of high quality, and the latent representations of documents are usually more discriminative. To show this point, we illustrate the topics learned by AVITM in Table 5 in Appendix 7.1 when K is set to 50. We can see that there are a lot of redundant topics. However, the topics learned by our model are diverse, which is shown in Table 4. We also show the TSNE of the latent representations of document learned by AVITM in Figure 4-(c) for comparison.

---

> ### Author Response · Authors · 2017-12-06
> **[Part-1] There might be some misunderstandings. We have addressed the misunderstandings and concerns of the reviewer.**
>
> We thank the reviewer for the valuable and detailed comments. The main concerns raised by the reviewer are:
> About the correctness of the model: Independently drawing the document-specific pi vectors from the stick-breaking processes will lead to zero sharing between the atoms of different stick-breaking process draws.
> About the novelty: The novelty of the paper is not significant.
> About the experiment details: The paper provides little information about how the two baselines, LDA and HDP, are implemented (e.g., via mean-field variational inference, VAE, or MCMC?) and how their perplexities and topic coherences are computed.
> About the learned topic numbers: It seems surprising that 20News dataset can be characterized by about 20 different topics.
>
> In this rebuttal, we have addressed all these concerns raised by the reviewer, and the paper has been modified accordingly.
>
> Q1. “Independently drawing the document-specific pi vectors from the stick-breaking processes will lead to zero sharing between the atoms of different stick-breaking process draws.”
>
> There might be some misunderstandings. In fact, the atoms of different stick-breaking process draws are shared in our model.  Actually, iTM-VAE does not need a base distribution to guarantee the sharing of topics across documents. Let us explain this point in 3 aspects:
>
> a. Why do traditional nonparametric Bayesian topic models, such as HDP, require a globally shared base distribution?
>
> Indeed, traditional nonparametric topic models, such as HDP, require a globally shared base distribution.  Note that, the main reason is that these models assume the “topics” are random variables and drawn from a distribution. As a result, people have to use a globally base distribution to generate a set of  countably infinite topics, such that these candidate topics are shared, otherwise the drawn topics will not be shared.  (c.f. Section 6.1 of [Teh et al. 2006], https://people.eecs.berkeley.edu/~jordan/papers/hdp.pdf).  Thus, a globally shared based distribution is used to generate a countably infinite set of topics that can be shared by different documents.
>
> b. Why the global base distribution is not required in iTM-VAE to make the atoms of different stick-breaking process draws shared by documents?
>
> Different with traditional nonparametric topic models, the topics in iTM-VAE are NOT drawn from a distribution, but are treated as part of the parameters that are optimized by the model directly. Specifically, in Section 4.1, we use $\Phi$ to denote the corresponding parameters. This key difference indicates that we do not need an additional base distribution to generate the countably infinite candidate topics, since they are parameters. Consequently, they are shared across all documents naturally.   Treating the topics as parameters are also adopted by LDA (c.f. the beta matrix in Section 3 of [Blei, et al. 2003] http://www.jmlr.org/papers/volume3/blei03a/blei03a.pdf), and the difference of our model with LDA is that the number of topics is potentially unlimited and  is adapted with the data.
>
> c. What if a nonparametric topic model does not share topic between documents?
>
> Another evidence that the model is correct is that, if the topics (atoms) are not shared by different stick-breaking processes, we cannot learn meaningful topics at all. However, the fact is that the topic coherence of our model is higher than other strong baselines. Please check the Section 7.1 in the revision to see the learned topics. We will release the code on github for you to check the correctness of the model. (The Section 7.1 in the revision corresponds to Section 7.3 in the original version. We add more experimental results and move it to the top of the Appendix such that readers can visualize the learned topics easily. )
>
> Thanks for this comment. We have added this discussion in the revision.

---

> ### Author Response · Authors · 2017-12-29
> **Dear Reviewer, we are still waiting for your feedback about whether our rebuttal and revision have addressed your concerns.**
>
> Dear Reviewer,
>
> Thanks very much for your review!
>
> We received the review at Dec 2nd and we have posted the rebuttal and the revision at Dec 6th. In the rebuttal, we explained why iTM-VAE does not need a globally shared sticking-breaking process and the documents DO shared the topics of the model. We also clarified the novelty of the paper and addressed some concerns of the reviewer.
>
> We hope that our rebuttal has addressed your concerns. And we are still waiting for your feedback. We are looking forward to hearing from you. Many thanks!

---

### Official Review · AnonReviewer1 · 2017-11-28
**seems to miss comparable non-deep comparisons**

**Rating:** 5
**Confidence:** 2

**Review:**

The paper proposes a VAE inference network for a non-parametric topic model.

The model on page 4 is confusing to me since this is a topic model, so document-specific topic distributions are required, but what is shown is only stick-breaking for a mixture model.

From what I can tell, the model itself is not new, only the fact that a VAE is used to approximate the posterior. In this case, if the model is nonparametric, then comparing with Wang, et al (2011) seems the most relevant non-deep approach. Given the factorization used in that paper, the q distributions are provably optimal by the standard method. Therefore, something must be gained by the VAE due to a non-factorized q. This would be best shown by comparing with the corresponding non-deep version of the model rather than LDA and other deep models.

---

> ### Author Response · Authors · 2017-12-06
> **We have compared our model with the non-deep counterpart method, and we have addressed all the concerns in the review.**
>
> We thank the reviewer for the valuable comments. The main concerns raised by the reviewer are:
> 1) About the model: There are no document-specific topic distributions, but only a stick-breaking process for a mixture model;
> 2) About the novelty: The difference with Wang, et al (2011) is that VAE is used to approximate the posterior. Hence the model should compare with Wang, et al (2011) since it is the most relevant non-deep approach.
>
> We address all these concerns raised the reviewer as follows:
> Q1: “The model on page 4 is confusing to me since this is a topic model, so document-specific topic distributions are required, but what is shown is only stick-breaking for a mixture model.”
>
> There might be some misunderstandings. We do have the document-specific topic distributions on page 4, which is $\pi$ in the generative process. The difference with LDA is that our model samples the document-specific topic distributions from a GEM distribution, while LDA samples them from a Dirichlet distribution. Actually, the generative procedure of iTM-VAE of Section 4.1 is similar to LDA, i.e. 1) sample a document-specific topic distributions $pi$; Then, for each word $w_i$ in the document: 2) draw a topic $\hat{\theta}_i$; 3) draw $w_i$ from $Cat(\hat{\theta}_i)$.
>
> Moreover, according to the section 3.2 of ( Blei2003, JMLR, http://www.jmlr.org/papers/volume3/blei03a/blei03a.pdf), LDA itself is also a mixture model.
>
> Q2. “In this case, if the model is nonparametric, then comparing with Wang, et al (2011) seems the most relevant non-deep approach.”, “This would be best shown by comparing with the corresponding non-deep version of the model rather than LDA and other deep models.”
>
> We agree with the reviewer that it would be best to compare with Wang, et al (2011) to see the gain from VAE. Actually, the HDP in Table 1 is taken from Miao, et al(2017), which is actually based on Wang, et al (2011). We have clarified this point in the revision. Thanks very much for the suggestion!
>
> Q3. About the novelty of the paper.
>
> We would like to clarify the novelty of the paper:
> 1)  As is pointed out by the reviewer, we introduce a global inference net to model the variational posterior of BNP topic models, and carry out optimization under the VAE framework. To our best knowledge, this is the first time that BNP topic models are combined with AEVB. This technique brings 2 benefits for BNP topic models. (1) No further variational updates are needed on the test data but only a feed-forward pass on inference net. Hence, the model is very efficient. (2) The optimization of VAE framework is very general. The generative model can be adjusted and enhanced without additional mathematic derivation. Hence, it is very flexible.
>
> 2) We propose iTM-VAE-G, in which a prior is added on the concentration parameter. This technique helps the model to adjust the concentration parameter to data automatically, and we have shown the effect of the prior in Section 5.4. To further demonstrate the advantage of iTM-VAE-G, we compare it with iTM-VAE-Prod which does not have the prior over the concentration parameter. This newly added experiments are shown in Table 2 in the revision. We can see that when the decoder is strong, the number of effective topics learned by iTM-VAE-Prod(without the prior) cannot adapt well to different sub-sampled subsets of 20News dataset. Common training techniques, e.g. KL annealing, decoder regularization, do not help much. iTM-VAE-G can increase the adaptive power of the model in an elegant way. After the prior is added, the restriction on the latent representation is relaxed, the model will learn an appropriate and highly-confident corpus-level posterior for the concentration parameter, and can adapt its power according to the dataset size, even if the decoder is strong. As commented by AR3, this technique is “very important and not well done in the community, usually”.
>
> 3) The experimental results confirm the advantage of the model.
>
> Please let us know whether the rebuttal solves the concerns of the reviewer. We are looking forward to hearing from you. Thanks very much!

---

> ### Author Response · Authors · 2017-12-29
> **Dear Reviewer, we are still waiting for your feedback about whether our rebuttal and revision have addressed your concerns.**
>
> Dear Reviewer,
>
> Thanks very much for your review!
>
> We received the review at Dec 2nd and we posted the rebuttal and the revision at Dec 6th. We hope that our rebuttal has addressed your concerns. And we are still waiting for your feedback. We are looking forward to hearing from you.

---

### Author Response · Authors · 2018-01-04
**We have revised our paper and we list the changes made to the original version according to the reviews.**

Dear Area Chair and Reviewers,

We have revised our submission according to your valuable comments. Here we list the changes made to the original submission for your convenience.

    -1. [AR1] In Table 1, we have clarified that the results of HDP are indeed based on Wang, et al (2011).

    -2. [AR2] In the last paragraph of Section 4.1 in the revision, we explain why our model does not need a global base distribution. And we also explain that the topics (atoms) do be shared across documents in different stick-breaking draws by iTM-VAE naturally. If, as AR2 worried, our model leads to “zero sharing” of topics, the model will learn nothing. In contrast, our model learns quite meaningful topics (c.f. Table 4 in the revision). We are confident that our method is theoretical correct. Please refer to Q1 of our rebuttal to AR2 for more details.

    -3. [AR2] In Section 5.4, we add a paragraph (the 2nd paragraph) to emphasize the novelty of iTM-VAE-G model, which imposes a prior over the concentration parameter, and analyze the benefits of iTM-VAE-G model. We also add an experiment to show the advantage of iTM-VAE-G over other commonly used tricks for VAE-based models to increase the adaptive power, such as KL annealing and regularizing the decoder in Table 2 and the 3rd and 4th paragraph of Section 5.4.

    -4. [AR2] In Section 7.1 of the revision, we provide more justifications on why iTM-VAE is better. We compare the sparsity and the TSNE of the representations of iTM-VAE and ProdLDA in Figure 4. We also list the topics learned by ProdLDA in Table 5. Compared to Table 4, there are a lot of redundant topics in Table 5.

    -5. [AR2 and AR3] In Table 1, we have clarified that the results of LDA and HDP are taken from Miao, et al. (2017). Since we use the exactly same datasets as Miao, et al. (2017), we can take the results of LDA and HDP directly. According to Miao et al. (2017), they use Hoffman et al., (2010) for LDA and Wang et al. (2011) for HDP, which are both based on variational inference.

    -6. [AR3] In the 3rd paragraph of Section 1, we have removed the claim “Inference of HDP is more complicated and not easy to be applied to new models …”

    -7. [AR3] In the 3rd paragraph of Section 2, we have removed "superior" and "self-determined topic number".

    -8. [AR3] In Related Work, we have added more references, such as Kim & Sudderth (2011); Archambeau et al. (2015) and Lim et al. (2016).

---

### Decision · Program_Chairs · 2018-01-29
**ICLR 2018 Conference Acceptance Decision**

**Decision:**

Reject

**Comment:**

The paper proposes a BNP topic model that uses a stick-breaking prior over document topics and performs VAE-style inference over them. Unfortunately, the novelty of this work is limited, as VAE-like inference for LDA-like models, inference with stick-breaking priors for VAEs, and placing a prior on the concentration parameter in a non-parametric topic model have all been done before (see e.g. Srivastava & Sutton (2017), Nalisnick & Smyth (2017), and Teh, Kurihara & Welling (2007) respectively). There are also concerns about the correctness of treating topics as parameters (as opposed to random variables) in the proposed model. The authors' clarification regarding this point was helpful but not sufficient to show the validity of the approach.